# Research and Fabrication of Broadband Ring Flextensional Underwater Transducer

**DOI:** 10.3390/s21041548

**Published:** 2021-02-23

**Authors:** Jiuling Hu, Lianjin Hong, Lili Yin, Yu Lan, Hao Sun, Rongzhen Guo

**Affiliations:** 1Acoustic Science and Technology Laboratory, Harbin Engineering University, Harbin 150001, China; zorro590@hrbeu.edu.cn (J.H.); honglianjin@hrbeu.edu.cn (L.H.); yinlili@hrbeu.edu.cn (L.Y.); sunhao12138@hrbeu.edu.cn (H.S.); guorongzhen@hrbeu.edu.cn (R.G.); 2Key Laboratory of Marine Information Acquisition and Security (Harbin Engineering University), Ministry of Industry and Information Technology, Harbin 150001, China; 3College of Underwater Acoustic Engineering, Harbin Engineering University, Harbin 150001, China; 4Shenyang LiaoHai Equipment CO., LTD., Shenyang 110000, China

**Keywords:** low frequency, broadband, flextensional transducer, flexural vibration, finite element

## Abstract

At present, high-speed underwater acoustic communication requires underwater transducers with the characteristics of low frequency and broadband. The low-frequency transducers also are expected to be low-frequency directional for realization of point-to-point communication. In order to achieve the above targets, this paper proposes a new type of flextensional transducer which is constructed of double mosaic piezoelectric ceramic rings and spherical cap metal shells. The transducer realizes broadband transmission by means of the coupling between radial vibration of the piezoelectric rings and high-order flexural vibration of the spherical cap metal shells. The low-frequency directional transmission of the transducer is realized by using excitation signals with different amplitude and phase on two mosaic piezoelectric rings. The relationship between transmitting voltage response (TVR), resonance frequency and structural parameters of the transducer is analyzed by finite element software COMSOL. The broadband performance of the transducer is also optimized. On this basis, the low-frequency directivity of the transducer is further analyzed and the ratio of the excitation signals of the two piezoelectric rings is obtained. Finally, a prototype of the broadband ring flextensional underwater transducer is fabricated according to the results of simulation. The electroacoustic performance of the transducer is tested in an anechoic water tank. Experimental results show that the maximum TVR of the transducer is 147.2 dB and the operation bandwidth is 1.5–4 kHz, which means that the transducer has good low-frequency, broadband transmission capability. Meanwhile, cardioid directivity is obtained at 1.4 kHz and low-frequency directivity is realized.

## 1. Introduction

Flextensional transducers (FTs) are typical low-frequency transducers which are usually used in the underwater acoustics, in which the flexural vibration of the shell is excited by piezoelectric ceramics [1,2,3,4]. Since Harvey C. Hayes from the U.S. Naval Research Laboratory successfully developed the first flextensional transducer in 1929 [5], seven structural forms of flextensional transducer have been developed over the past several decades [6,7]. In the seven structural forms, the class V flextensional transducer adopt mosaic piezoelectric ring, which is constructed by wedge-shaped piezoelectric ceramic strips, to drive the two spherical cap metal shells to achieve low frequency sound. In 1984, G.W. McMahon established the theoretical system of the class V flextensional transducer [8]. Then, in 1987, G.W. McMahon succeeded in developing a class V flextensional transducer, with a resonance frequency of 600 Hz, the output acoustic power of which is 10 kW. However, its –3 dB bandwidth is 160 Hz [9].

The class V flextensional transducer exhibits excellent low frequency performance [10,11]. However, the spherical cap metal shells driven by mosaic piezoelectric rings only have fundamental flexural vibration mode. Therefore, the transducer just works at a single resonance frequency with a narrow work bandwidth and it is also limited in underwater acoustic communication equipment which needs broadband performance of the transducer [12,13,14]. Meanwhile, since its ka value is far less than 1, the class V flextensional transducer has omnidirectional radiation [15,16], which is unfavorable for anti-reverberation and anti-multipath in shallow water, and also seriously reduces its application value.

This paper proposes a new type of ring flextensional underwater transducer which is composed of two mosaic piezoelectric rings and two spherical cap metal shells. The transducer changes the spherical cap flexural vibration mode used in the class V flextensional transducer to the flexural vibration mode of the mosaic ring at the low-frequency. Because of the vibration mode changes, the working bandwidth of the transducer is expanded by coupling the second order flexural vibration mode of the spherical cap metal shells and the low-frequency vibration mode of the transducer. Meanwhile, the excitation control of two mosaic piezoelectric rings is utilized to excite the modes of monopole and dipole simultaneously. The low-frequency direction of the transducer is achieved by combine with monopole and dipole sound pressure. In Section 2 and Section 3 of this paper, the electroacoustic characteristics simulation and optimization of the transducer by the finite element method are introduced emphatically. Then, the broadband ring flextensional underwater transducer is designed and fabricated. Finally, the test results of the transducer are presented and compared with the results of finite element analysis. The test results indicate that the ring flextensional underwater transducer has the excellent low frequency, broadband and low-frequency direction performance.

## 2. The Influences of Structural Parameters of the Transducer

The schematic structure of the transducer is shown in Figure 1. The transducer consists of upper and lower spherical cap aluminum alloy shells, multiple groups of Lead Zirconate Titanate (PZT) bars and one ring base. Each group of PZT bars is composed of one PZT bar which is polarized clockwise and one PZT bar which is polarized anticlockwise. The PZT bars are inlaid in the reserved holes of spherical cap aluminum alloy shells. They are combined to radiate the acoustic power to the water. The ring base plays a role of keeping the distance between the upper and lower shells. The polyurethane rubber layer is filled between the aluminum alloy shells and ring base in order to restrain unnecessary coupling vibration between shells and ring base. The structural parameters of the transducer are shown in Table 1 and PZT4 is selected as piezoelectric material.

As shown in Figure 2, The 3-D finite element model was used to calculate the transmitting characteristics. Considering the symmetric geometrical structure of the transducer, 1/8 models were established, and symmetric boundary conditions were applied to improve the simulation efficiency. A water domain with a radius of 600 mm was constructed around the transducer, the outer surfaces of the water were covered with a perfectly matched layer (PML) to prevent the reflection of the waves radiated by the transducer. The dimension of each element in the water domain was less than 1/6 of the wavelength, and such a mesh was implemented to achieve sufficient accuracy and stability of results. The type of medium filling the transducer was air, whose mass density and sound speed are 1.29 kg/ m^3^ and 343 m/s, respectively. The properties of Aluminum and PZT-4 used in the simulation can be found in Appendix A and Table A1. The upper and lower PZT ring were driven independently by two electronic circuits. Furthermore, the PZT bars that make up the upper PZT ring are connected in parallel, as with the lower PZT ring.

Under the setting of basic parameters, the transmitting voltage response (TVR) curve of the transducer is obtained through the finite element analysis, as shown in Figure 3. The TVR of the transducer shall be always maintained above 140 dB in the 1.4–5 kHz region and the fluctuation in the band is less than 6 dB. Meanwhile, there are two vibration modes, one is at 2.0 kHz and other at 3.8 kHz. These two modes play the major contributions to the TVR in the horizontal direction. Respectively, the TVR is 146.0 dB at 2.0 kHz and 145.8 dB at 3.8 kHz.

Based on the modal analysis, the two vibration modes of transducer were obtained, as shown in Figure 4a,b. These modes were denoted as the first vibration mode and the second vibration mode. The resonance frequency of the first vibration mode was 2.86 kHz and the vibration mode was similar to the breathing mode of the PZT ring. However, since the upper boundary of the PZT ring was restricted by the metal shells, the first vibration mode was manifested as radial vibration of the lower part of the PZT ring, accompanied by slight flexural vibration of the metal shells. The resonance frequency of the second vibration mode was 4.8 kHz. The vibration form was reflected in the flexural vibration of the metal shells, accompanied by a slight swing of the PZT ring.

The paper studies the effects of parameter variation of the transducer on underwater resonance frequency of each vibration modes and TVR. The first influence factor was the diameter of the PZT ring. Figure 5b shows the effect of changes in diameter of the PZT ring on resonance frequencies of the first and second vibration modes in water. The solid line is the frequency variation curve of the first vibration mode and the dashed line is the frequency variation curve of the second vibration mode. With a gradual increase in the diameter of the PZT ring, the resonance frequencies of the first and second vibration modes gradually decrease. The frequency decrease of the first vibration mode is attributed to the effect of the diameter of the PZT ring on the frequency of the breathing mode of the PZT ring. The greater the diameter, the lower the frequency for the resonance frequency of first vibration mode [17,18,19]. The reason for the frequency decrease of the second vibration mode is an increase in the diameter of the metal shells with an increase in the diameter of the PZT ring. The flexural vibration is affected by the diameter of the metal shells. The greater the shells diameter, the lower the resonance frequency for the flexural vibration.

Figure 5c shows the effect of changes on the diameter of the PZT ring on the maximum values in TVR of the first and second vibration modes. With a gradual increase in the diameter of the PZT ring, the peak TVR of the first vibration mode shows a linear change and gradually decreases. The peak TVR of the second vibration mode attenuates at a low rate at first, but then the attenuation accelerates when the diameter is beyond 360 mm. Therefore, when the diameter of the transducer is 360 mm, the difference between the underwater peak TVR of two vibration modes is the minimum.

Second, the paper studies the effect of changes in the height of the PZT bars on underwater resonance frequency of the transducer, as shown in Figure 6b. The effect of changes in the height of the PZT bars on the peak TVR of the transducer is shown in Figure 6c, in which the solid line shows changes in the first vibration mode and the dashed line shows changes in the second vibration mode. Figure 6b,c indicate that as the height of the PZT bars gradually increases, the resonance frequency of the first vibration mode shows a slight downward trend. But the peak TVR gradually increases, showing a minor effect of changes in the height on the first vibration mode. However, since more active materials are used, the acoustic power that the transducer radiates increases and the TVR increases. Meanwhile, with gradual increase of the height of the PZT bars, the resonance frequency of the second vibration mode presents a trend of linear decrease and its TVR still increases gradually with the use of more active materials.

The decreasing rate of the resonance frequency of the second vibration mode is faster than that of the first vibration mode. Therefore, with the increase of the height of the PZT bars, the difference of the resonance frequencies of the two vibration modes gradually becomes smaller. When the height is more than 80 mm, the difference between resonance frequencies of two vibration modes are less than 1.5 kHz. Meanwhile, with an increase in height, the difference between the peak TVR of the two vibration modes gradually increases. Furthermore, when the height of the PZT bars is more than 90 mm, the TVR of the second vibration mode is more than 1 dB more than that of the first vibration mode.

Finally, the effect of the height of the ring base on underwater resonance frequency of the transducer was studied. As shown in Figure 7b, with an increase in the height of the ring base, frequency of the first and second vibration modes show minor change. Because the frequency of first vibration mode is mainly affected by the parameters of PZT ring, and the frequency of the second vibration mode is mainly affected by the parameters of the metal shell, increasing the height of the base has less effect on them. Therefore, the change in the height of the ring base has less effect on the frequency of the two vibration modes. The effect of the height of the ring base on the peak TVR is shown in Figure 7c. With an increase in the height, the change in the peak TVR of the first vibration mode is not obvious. The peak TVR of the second vibration mode gradually increases first and then it remains unchanged. However, when the height of the ring base is more than 50 mm, the difference between the peak TVR of the two vibration modes is less than 1 dB.

## 3. Synthetic Directivity

The previous sections analyze the effects of changes in the parameters of the transducer on resonance frequency and transmitting voltage response, so as to achieve the operation of bandwidth and TVR. However, in underwater acoustic communication or underwater detection, sometimes in order to avoid interference of multipath, the transducer will have directionally radiating capability [20,21], such as “Cardioid” directivity, which can be formed by superposition of one omnidirectional sound source and one sound source with the dipole radiation characteristic [22,23,24,25], as shown in Figure 8.

This section elaborates on the principle of the flextensional transducer made of PZT rings and metal shells to radiate sound wave directionally. The directivity of the transducer is closely related to the working frequency of the transducer. In order to realize the “Cardioid” directivity characteristic and high-power radiation in low frequency, the frequency of the transmitting “Cardioid” directivity is selected as 1.4 kHz by comparing the TVR response of the monopole excitation mode with that of dipole mode.

First, the transducer was divided into two parts and each part contained one PZT ring and one metal shell, as shown in Figure 9. The upper part was applied with an excitation signal with V_1_ amplitude and 0 phase at 1.4 kHz, which was denoted as excitation signal +V_1_. The lower part is applied with the same excitation signal +V_1_. The directivity pattern of the transducer is as shown in Figure 10, which shows that the transducer is omnidirectional in a vertical direction. In the same frequency, the upper part is applied with excitation signal +V_2_ and the lower part is applied with the same amplitude V_2_, but the phase is 180° different to the excitation signal that applied to the upper part. The directivity of the transducer is in the form of dipole, as shown in Figure 11.

As shown in Figure 8, in order to achieve the “Cardioid” directivity, sound pressure weights of monopole and dipole need to be the same. Therefore, the amplitude of V_2_ shall guarantee that the acoustic pressure of dipole directivity and omnidirectional directivity are equal in the 90° direction in the far field. Now the excited signals were redistributed in upper and lower parts of the transducer, in which the excited signal to the upper part is +V_1_+V_2_ and the excited signal to the lower part is +V_1_–V_2_, as shown in Table 2. Under this condition, the directivity pattern is shown in Figure 12, which is close to a “Cardioid”. The difference between acoustic pressure levels at the 90° position and 270° position is more than 10 dB, satisfying the unidirectional transmitting requirement.

## 4. Experiment and Test

The actually fabricated components of the transducer are shown in Figure 13a, geometric error of the finished components is less than 5%. The prototype of transducer is shown in Figure 13b. Basic parameters of the transducer are the same as those shown in Table 1. The diameter is 360 mm, the integral height is 300 mm and the surface of the transducer is covered by polyurethane rubber to guarantee the waterproof property of the transducer.

The admittance characteristics of the transducer in water are tested using a WK6100 impedance analyzer, and the measuring results are shown in Figure 14, which is compared with simulation results. Within the frequency range of 1.0–5.0 kHz, measured results are in reasonable agreement with simulated results. Comparison of the resonance frequency is shown in Table 3, the resonance frequency of the first vibration mode simulated in water is 2.0 kHz, and the experimental result is 2.2 kHz. The resonance frequency of the second vibration mode simulated in water is 3.7 kHz, and the actually tested frequency is 3.5 kHz, the error is less than 10%.

The TVR and directivity characteristic of the transducer were actually tested in Qiandao Lake, Zhejiang province, China. At the test site, the depth of the lake was 72 m. The schematic diagram of the experimental process is shown in Figure 15. The transducer was located at 30 m below the lake water surface and the hydrophone was located at the same depth. The distance between transducer and hydrophone was 5 m. Here we used two JYH 2000 A single-channel power amplifiers with up to 50 dB continuous variable gain, 50 Hz to 40 kHz working bandwidth, to drive the upper and lower part of the transducer independently and apply different voltages. The output voltages of the power amplifiers were controlled by gain tuning and monitored by oscilloscope through the –40 dB attenuated monitoring ports of the power amplifiers. The signals were received on a B&K hydrophone (type 8105), which is a spherical transducer for making absolute sound measurements over the frequency range 0.1 Hz to 160 kHz, whose free-field voltage response at 4 kHz was calibrated at –207.6 dB re 1 V/μPa. The received signal was filtered and amplified by an NF 3628 programmable filter, using its narrowband bandpass mode. A comparison was made of the actually measured TVR and simulation TVR in 1.0–5.0 kHz, and the results are shown in Figure 16. The error between them is less than 5%. Comparison of the peak TVR is shown in Table 4. The peak TVR of first vibration mode is 146.9 dB and the corresponding frequency is 1.9 kHz. The peak TVR corresponding to the second vibration mode is 147.2 dB, and the corresponding frequency is 3.7 kHz. The error compared with the results of simulation is less than 5%.

The directivity characteristic of the transducer was tested in the same measurement condition. Based on the analysis in Table 2, the amplitude of the acoustic pressure which is gained by applying +1.3 V_0_ voltage to the upper part and –1.3 V_0_ voltage to the lower part at 1.4 kHz is equal with the amplitude of the acoustic pressure which is gained by applying +V_0_ voltage to the upper and lower parts simultaneously at 1 m in the 90° direction. Therefore, in order to obtain the monopole and dipole mode of the transducer, +2.3 V_0_ excited voltage needs to be applied to the upper part, and –0.3 V_0_ to the lower part, in which V_0_ is a constant. The direction of the projector was adjusted and controlled by step controller. The receiving terminal records test data once every 2° each time the transducer rotates. Comparing the directivity obtained in measurements and the finite element, the result is shown in Figure 17. It can be seen that the transducer mainly radiates acoustic energy toward the 90° direction and less in other directions. The difference between acoustic pressure levels at 90° position and 270° position is more than 10 dB. The directional emission of the transducer is achieved.

## 5. Conclusions

This paper proposes a new ring type of flextensional transducer constructed of double piezoelectric rings and spherical cap shells. By contrast with previous work, this transducer works in a broad band with a higher source level. Meanwhile, it has better directivity and will be more favorable to use for underwater acoustic communication or underwater detection. The working band of the transducer is expanded by coupling the breathing mode of the rings with the flexural mode of the spherical cap shells. The finite element simulation shows that the TVR of the transducer is more than 140 dB in 1.4–5 kHz and the fluctuation of TVR is less than 6 dB. The perfect consistency is shown in comparison between the actual test results and finite element simulation. Meanwhile the paper achieves the Cardioid directional transmission of the transducer at 1.4 kHz based on the mechanism of synthetic directivity. Therefore, the transducer will have wide application in underwater communication and detection. For the limitation of air backing structure in pressure tolerance, the free flooded structure of the ring flextensional transducer can be constructed by punching holes on the ring base to increase the operating depth of the transducer significantly. The effect of the free flooded structure transducer will be one of the contents to be studied next.

## Figures and Tables

**Figure 1 sensors-21-01548-f001:**
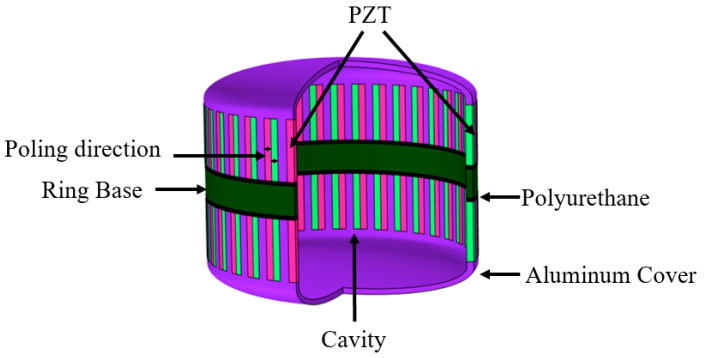
Schematic structure of the broadband ring flextensional underwater transducer.

**Figure 2 sensors-21-01548-f002:**
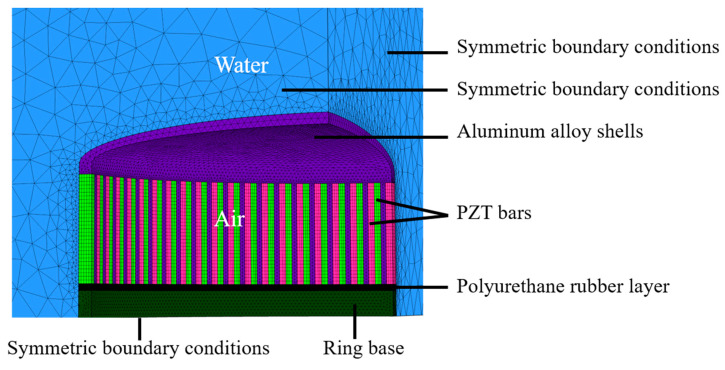
The finite element model of the transducer.

**Figure 3 sensors-21-01548-f003:**
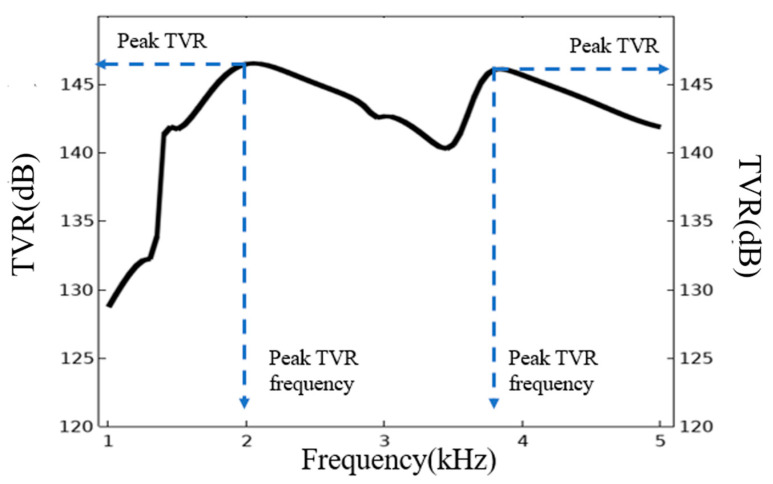
The transmitting voltage response (TVR) curve of the transducer with the basic dimensions.

**Figure 4 sensors-21-01548-f004:**
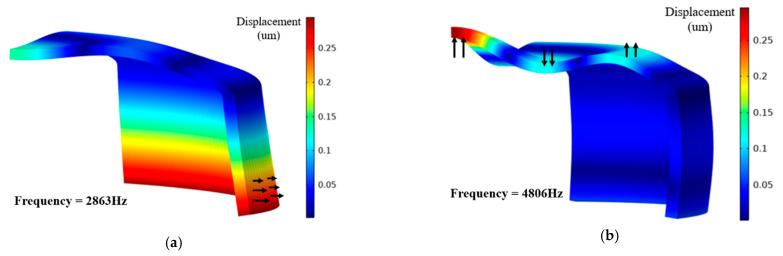
(**a**) Displacement of the first vibration mode at low frequency which is mainly a radial vibration of the lower part of PZT ring. (**b**) Second vibration mode shape which is mainly the high order flexural vibration mode of the metal shells.

**Figure 5 sensors-21-01548-f005:**
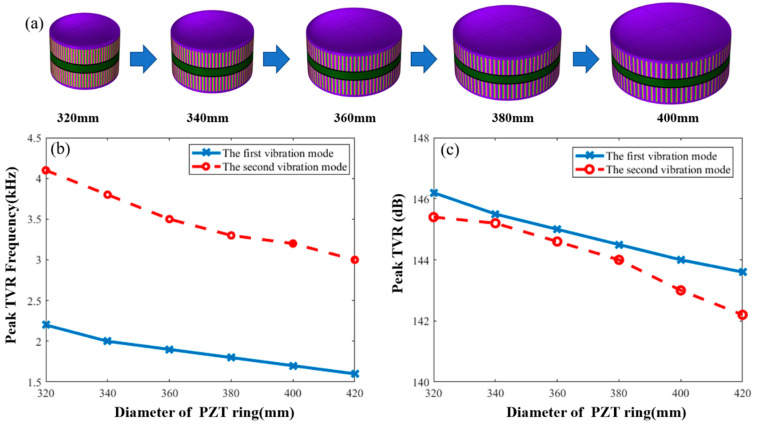
(**a**) Macroscopic shape changes of transducers. (**b**) Effect of the PZT ring diameter on frequency that is corresponding to the peak TVR. (**c**) Effect of the PZT ring diameter on peak TVR value for first and second vibration modes.

**Figure 6 sensors-21-01548-f006:**
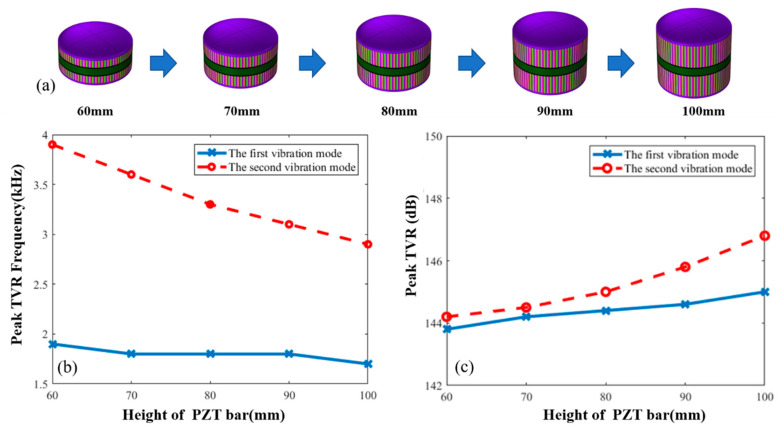
(**a**) Macroscopic shape changes of transducers. (**b**) Effect of the PZT bar height on frequency that is corresponding to the peak TVR. (**c**) Effect of the PZT bar height on peak TVR value for first and second vibration modes.

**Figure 7 sensors-21-01548-f007:**
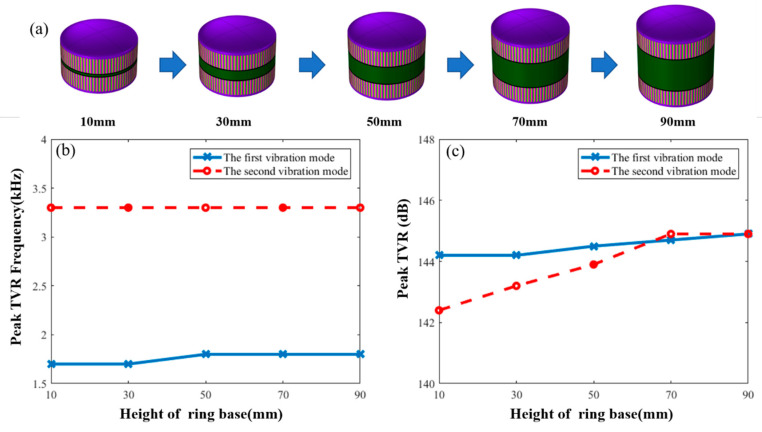
(**a**) Macroscopic shape changes of transducers. (**b**) Effect of the ring base height on frequency that is corresponding to the peak TVR. (**c**) Effect of the ring base height on peak TVR value for first and second vibration modes.

**Figure 8 sensors-21-01548-f008:**
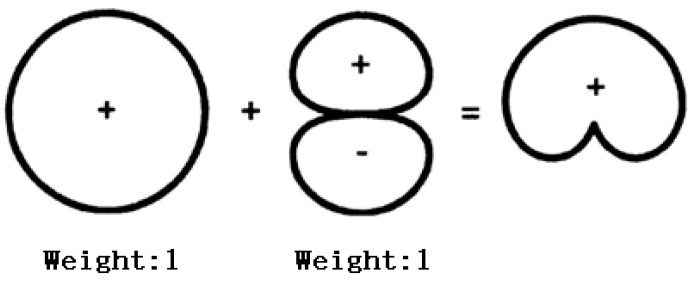
Schematic diagram of synthetic directivity.

**Figure 9 sensors-21-01548-f009:**
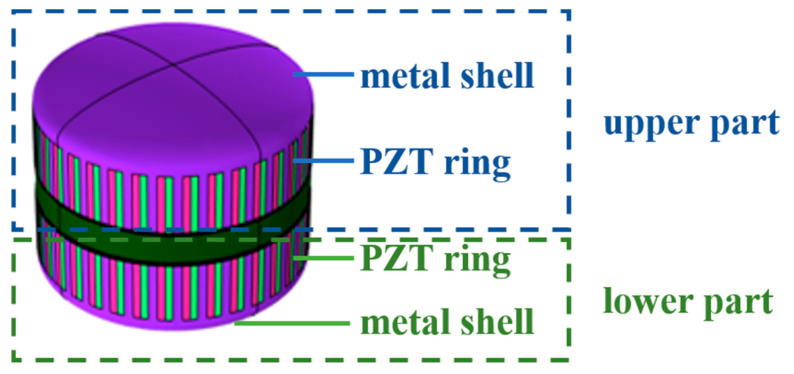
Segmentation diagram of the transducer for excitation. each part includes one PZT ring and one metal shell.

**Figure 10 sensors-21-01548-f010:**
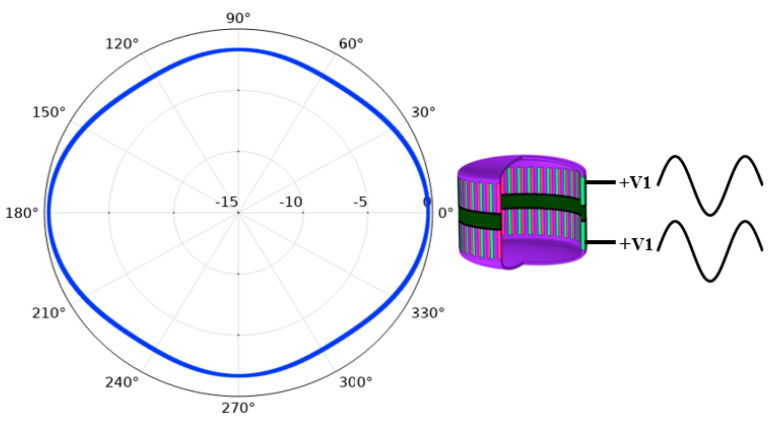
Vertical directivity pattern of the transducer when the same excitation signals are applied to the upper and lower parts.

**Figure 11 sensors-21-01548-f011:**
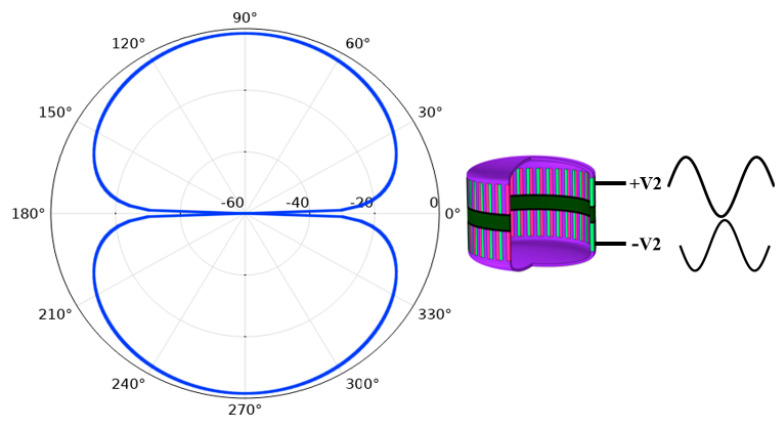
Vertical directivity pattern of the transducer when the excitation signals’ phase difference applied to the upper and lower parts is 180 degrees.

**Figure 12 sensors-21-01548-f012:**
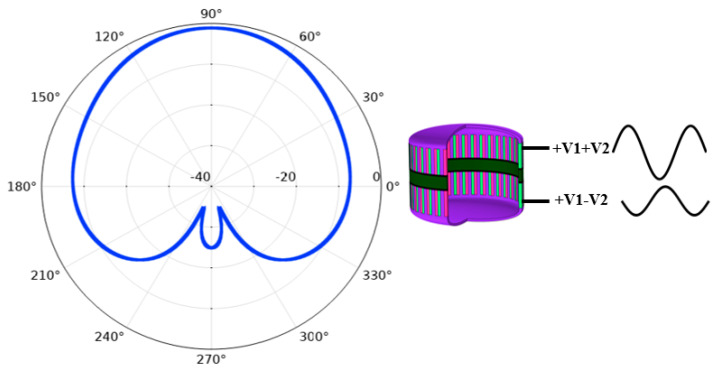
Vertical Cardioid directivity of the transducer.

**Figure 13 sensors-21-01548-f013:**
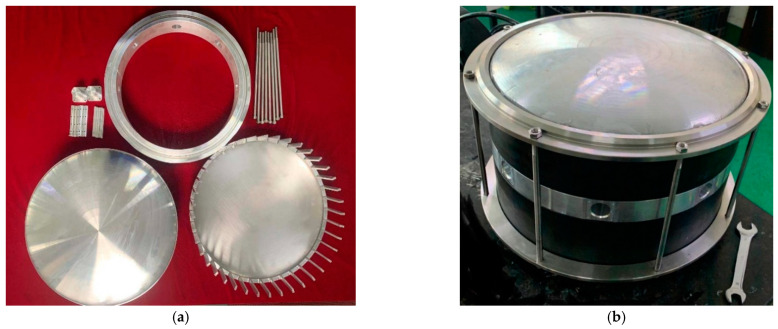
(**a**) Assembling components of transducer. (**b**) Fully assembled transducer.

**Figure 14 sensors-21-01548-f014:**
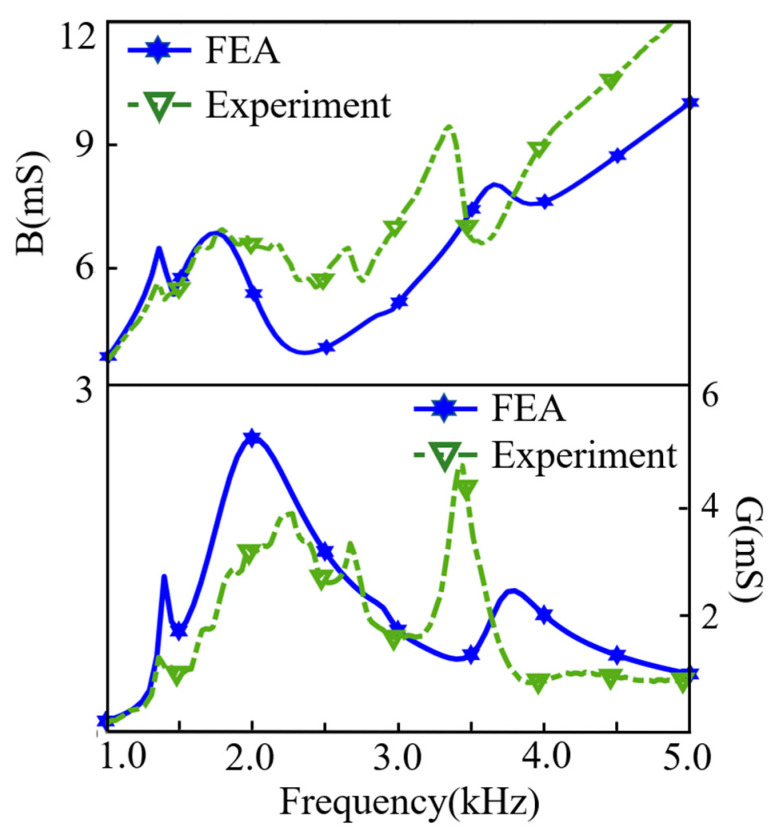
Comparison of the underwater impedance response between the Finite Element Analysis (FEA) and measurement.

**Figure 15 sensors-21-01548-f015:**
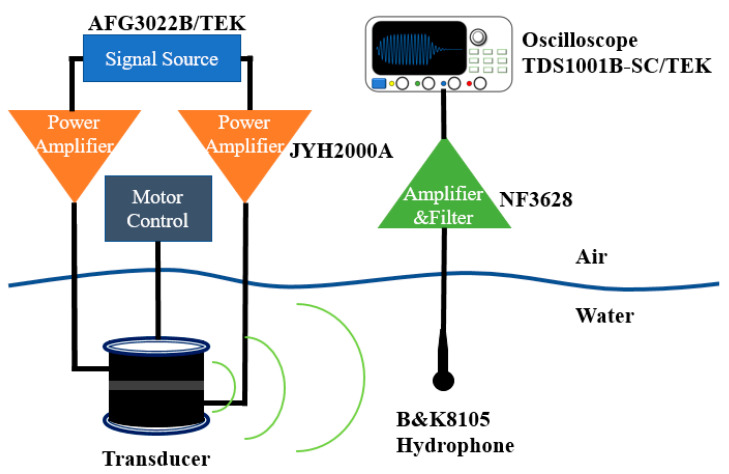
Block diagram of the experimental system.

**Figure 16 sensors-21-01548-f016:**
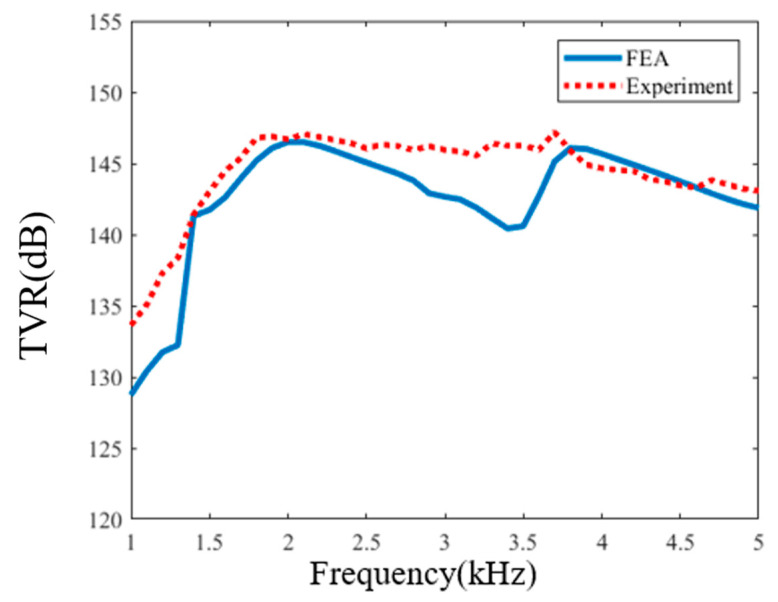
Comparison of the TVR curve from the FEA and the measurement.

**Figure 17 sensors-21-01548-f017:**
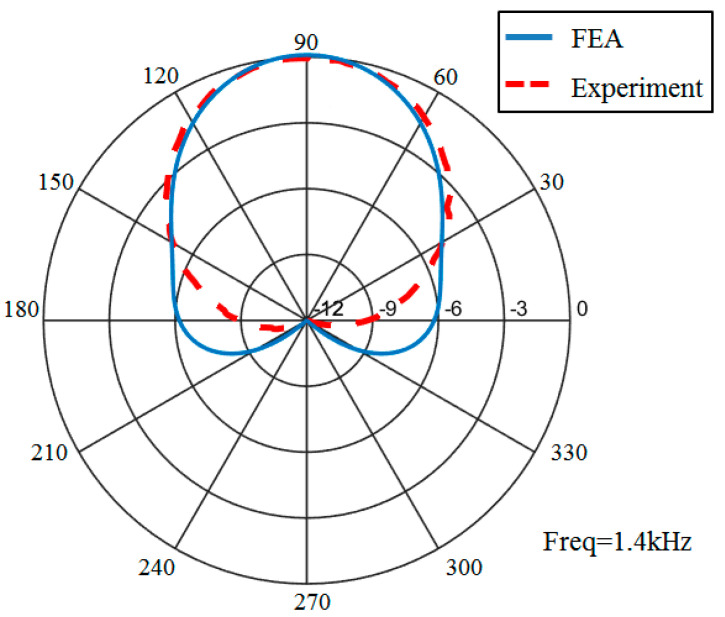
Comparison of the directivity from the FEA and the measurement.

**Table 1 sensors-21-01548-t001:** Structural parameters of the ring flextensional underwater transducer.

Parameter	Basic Dimension(mm)
PZT bar thickness	10.0
PZT bar length	5.0
PZT bar height	80.0
PZT ring diameter	360.0
Aluminum cover thickness	8.0
Aluminum cover height	23.0
Ring base height	40.0
Polyurethane height	8.0

**Table 2 sensors-21-01548-t002:** The amplitude of the excitation signal for different vertical dipole directivity.

Directivity	Frequency (kHz)	Acoustic Pressure(dB)	Excited Signal(upper)	Excited Signal(lower)
Monopole	1.4	141.3	+V_1_	+V_1_
Dipole	1.4	139.0	+V_2_ = +1.3V_1_	−V_2_ = −1.3V_1_
Cardioid	1.4	147.3	2.3V_1_	−0.3V_1_

**Table 3 sensors-21-01548-t003:** Resonance frequency comparison for the two modes.

Resonance Frequency	FEA	Measurement
The first vibration mode (kHz)	2.0	2.2
The second vibration mode (kHz)	3.7	3.5

**Table 4 sensors-21-01548-t004:** Comparison of the peak TVR frequency and level from the FEA and the measurement.

Characteristic		FEA	Measurement
The First Vibration Mode	Peak TVR frequency(kHz)	2.0	1.9
	Peak TVR level(dB)	146.0	146.9
The Second Vibration Mode	Peak TVR frequency(kHz)	3.8	3.7
	Peak TVR level(dB)	146.4	147.2

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
