# Peer review of "Research and Fabrication of Broadband Ring Flextensional Underwater Transducer"

_sensors, 2021, doi:10.3390/s21041548_

Round 1

Reviewer 1 Report

Dear authors,

The reviewer has no doubts about the relevance of your research. The results of the work are of great interest to specialists in the field of underwater acoustics.

However, it is worth noting some drawbacks.

The conditions for conducting experimental studies are described too briefly. For example, according to the reviewer, the following questions are of interest:

  1. What is the bandwidth of the power amplifiers used in the radiation path? What is the gain? The authors give their brand, but the reader may be interested in this information as a source of information in the process of getting acquainted with the article. It is worth mentioning this very briefly.
  2. A similar question arises regarding the hydrophone used in the experiment. Its brand is given. However, a very brief mention of its data is of interest: the type of transducer; the resonance and the range of operating frequencies (in comparison with the frequency of its resonance); the unevenness of the frequency response of sensitivity in the range of operating frequencies, etc.
  3. A very brief mention of the frequency properties of the amplifier and filter connected to the output of the hydrophone may also be of some interest to the reader. Here, too, it is not enough to limit ourselves to just mentioning the brand of the equipment.

The need to supplement the materials of the article with answers to the questions listed in paragraphs 1-3 is of a recommendatory nature. The authors should independently assess the necessity of fulfilling this wish. The authors should take into account that if there is a significant interest in their work, it may be necessary to reproduce the results of the research. In this regard, the indication of the mentioned data is desirable for the successful reproduction of the results. It is important for followers to present the considerations that guided the authors when selecting the measuring equipment.

Besides:

  1. The article specifies the depth of the transducer and the hydrophone, but does not specify the distance to the bottom of the reservoir at the test site. If this distance is small, then it is possible that the results are incorrect. In any case, it is worth specifying this issue.
  2. It would be useful to include in the article the circuit of the transducer in the section, taking into account the electrical connection of the piezoelectric elements to each other.
  3. It is probably worth paying more attention to the description of the process of building a transducer model in the Comsol program. The paper does not describe the principles on which the selection of the grid element size is based when dividing the transducer into finite elements. The basic boundary conditions at the interface of the media, as well as the type of medium filling the transducer, are not described. It is desirable to say about the type of input electrical action on the transducer, etc.
  4. You need to correct typos: The @ sign is present in the axis designation in Fig. 2. Similarly in Fig. 5, 6, 15.

Author Response

Dear reviewer,

We wish to thank you for the time and effort you have spent reviewing our manuscript entitled “Research and Fabrication of Broadband Ring Flextensional Underwater Transducer” (ID: 1112444).

Motivated by your comments, we have deeply reconsidered the contents of our work and tried to fix all the points you mentioned. In particular, the revised manuscript has been improved significantly (The improvements were coloured in blue in the revised version for your convenience), and the comments and responses are summarized as follows:

Point 1: What is the bandwidth of the power amplifiers used in the radiation path? What is the gain? The authors give their brand, but the reader may be interested in this information as a source of information in the process of getting acquainted with the article. It is worth mentioning this very briefly.

Response 1:  Some details of the power amplifiers are added in the manuscript. (Page 10. Line 243-247) In addition, the power amplifier we used in the test is a general type of power amplifier designed for underwater acoustic transducers, especially the piezoelectric transducers. We controlled the output voltage of the power amplifiers by fine turning the continuous variable gain knobs, and monitored its voltages to make sure the voltage applied on the transducer is 200Vpp in the TVR test, and 230Vpp and 30Vpp respectively in the directivity test. As the transducer under these excitations is working in the linear range, the applied voltages are just intermediate variable, which would not influence the accuracy of the test results.

Point 2: A similar question arises regarding the hydrophone used in the experiment. Its brand is given. However, a very brief mention of its data is of interest: the type of transducer; the resonance and the range of operating frequencies (in comparison with the frequency of its resonance); the unevenness of the frequency response of sensitivity in the range of operating frequencies, etc.

Response 2: Some details of the hydrophone are added in the manuscript. (Page 10. Line 247-250) The hydrophone we used in the test has a very flat frequency response of sensitivity in the range of 0.5-10kHz.

Point 3: A very brief mention of the frequency properties of the amplifier and filter connected to the output of the hydrophone may also be of some interest to the reader. Here, too, it is not enough to limit ourselves to just mentioning the brand of the equipment.

Response 3: Some details of the filter are added in the manuscript. (Page 10. Line 250-251)

Point 4: The article specifies the depth of the transducer and the hydrophone, but does not specify the distance to the bottom of the reservoir at the test site. If this distance is small, then it is possible that the results are incorrect. In any case, it is worth specifying this issue.

Response 4: In the manuscript, the depth of the lake at the test site is added. (Page 10. Line 240)

Point 5: It would be useful to include in the article the circuit of the transducer in the section, taking into account the electrical connection of the piezoelectric elements to each other.

Response 5: The circuit of the transducer including electrical connection of the piezoelectric elements are added in the manuscript. (Page 3. Line 89-91)

Point 6: It is probably worth paying more attention to the description of the process of building a transducer model in the Comsol program. The paper does not describe the principles on which the selection of the grid element size is based when dividing the transducer into finite elements. The basic boundary conditions at the interface of the media, as well as the type of medium filling the transducer, are not described. It is desirable to say about the type of input electrical action on the transducer, etc.

Response 6: The COMSOL modal is further introduced in the manuscript, the geometry, material settings, boundary conditions and the mesh types are explained in Figure 2. (Page 3. Line 80-93)

Point 7: You need to correct typos: The @ sign is present in the axis designation in Fig. 2. Similarly in Fig. 5, 6, 15.

Response 6: The axis designation is expressed in another style in this manuscript. (Page 4. Line 101, Page 5. Line 127, Page 6. Line 156, 171, Page 11. Line 259)

We look forward to hearing from you regarding our submission. We would be glad to respond to any further questions and comments that you may have. We appreciate for your warm work earnestly, thank you once again for your comments and suggestions.

Sincerely,

Corresponding author: Jiuling Hu, Yu Lan

Reviewer 2 Report

Authors proposed new broadband ring flextensional underwater transducer using PZT materials based double mosaic piezoelectric ceramic rings and spherical cap metal shells. Authors showed the experimental results of voltage responses vs. frequency curve graphs which are stable performances. FEA and experimental data could be quite similar with small errors. Therefore, authors showed good research work with new type of the transducers. 

However, authors do not compare the previous research in the literatures so authors need to provide the previous research in detail. In addition, authors need to emphaize the novelty in the conclusion section. Authors need to use professional English service or ask native English colleagues to improve the manuscript because there are some broken English expression. Except those major comments above, authors need to answer the following comments. Therefore, the manuscript can be improved if authors answer the following comments.

1. In Line 104, figure 4 -> Figure 4.

2. Authors need to add the reference (The greater the diameter, the lower the frequency for the resonance frequency of first ~).

3. Figure 4(c)shows -> Figure 4(c) shows

4. Authors need to improve Figure 8 quality.

5. Figure 5(c)indicate -> Figure 5(c) indicate

6. Authors need to explain (change in the height of the ring base has less effect on the two vibration modes ~). Is this why ?

Author Response

Dear reviewer,

We wish to thank you for the time and effort you have spent reviewing our manuscript entitled “Research and Fabrication of Broadband Ring Flextensional Underwater Transducer” (ID: 1112444).

Motivated by your comments, we have deeply reconsidered the contents of our work and tried to fix all the points you mentioned. In particular, the revised manuscript has been improved significantly (The improvements were coloured in blue in the revised version for your convenience), and the comments and responses are summarized as follows:

Point 1: However, authors do not compare the previous research in the literatures so authors need to provide the previous research in detail.

Response 1:  In the revised version, the bandwidth of McMahon’s class V flextensional transducer is added in the first introduction paragraph. (Page 1. Line 44)

Point 2: In addition, authors need to emphasize the novelty in the conclusion section.

Response 2: The novelty of this research is summarized in the conclusion part. (Page12. Line 278-280)

Point 3: Authors need to use professional English service or ask native English colleagues to improve the manuscript because there are some broken English expression.

Response 3: We used professional English service to improve the English of this manuscript.

Point 4: In Line 104, figure 4 -> Figure 4.

Response 4: In the manuscript, the capital / small letter errors are fully checked and corrected (Page 4. Line 116)

Point 5: Authors need to add the reference (The greater the diameter, the lower the frequency for the resonance frequency of first ~).

Response 5: In the manuscript, how the diameter of the PZT ring and the height of the ring base influence the resonance frequency of the first and second vibration mode is further explained. (Page 6. Line 162-165)

Point 6: Figure 4(c)shows -> Figure 4(c) shows

Response 6: The spaces between words are fully checked and corrected. (Page 5. Line 132)

Point 7: Authors need to improve Figure 8 quality.

Response 6: The quality of this figure is improved by sharpen the font type. (Page 7. Line 199)

Point 7: Figure 5(c)indicate -> Figure 5(c) indicate

Response 7: The spaces between words are fully checked and corrected. (Page 5. Line 142)

Point 8: Authors need to explain (change in the height of the ring base has less effect on the two vibration modes ~). Is this why ?

Response 8: In the manuscript, how the diameter of the PZT ring and the height of the ring base influence the resonance frequency of the first and second vibration mode is further explained. (Page 6. Line 162-165)

We look forward to hearing from you regarding our submission. We would be glad to respond to any further questions and comments that you may have. We appreciate for your warm work earnestly, thank you once again for your comments and suggestions.

Sincerely,

Corresponding author: Jiuling Hu, Yu Lan